# Prevalence and the association between clinical factors and Diabetes-Related Distress (DRD) with poor glycemic control in patients with type 2 diabetes: A Northern Thai cross-sectional study

Achiraya Ruangchaisiwawet[1], Narumit Bankhum[2], Krittai Tanasombatkul[1,3], Phichayut Phinyo[1,3,4], Nalinee Yingchankul[1]*

1 Department of Family Medicine, Faculty of Medicine, Chiang Mai University, Chiang Mai, Thailand,
2 Nutrition and Dietary service section, Faculty of Medicine, Chiang Mai University, Chiang Mai, Thailand,
3 Center for Clinical Epidemiology and Clinical Statistics, Faculty of Medicine, Chiang Mai University, Chiang Mai, Thailand, 4 Musculoskeletal Science and Translational Research (MSTR), Chiang Mai University, Chiang Mai, Thailand

* nalineey@hotmail.com

## Abstract

### Background

Glycemic control is important to prevent diabetic complications. However, evidence linking factors such as diabetes-related distress (DRD) to poor glycemic outcomes is lacking in Thailand. Therefore, this study aimed to investigate the prevalence and associated factors of poor glycemic control type 2 diabetes.

### Methods

A cross-sectional study was conducted on 127 type 2 diabetic patients between December 2021 and March 2022 at Maharaj Nakorn Chiang Mai Hospital, Thailand. Data collection included demographic data, clinical data (duration of being type 2 diabetes, diabetic treatment modalities, weight, height, blood pressure, FBS, and HbA1c), behavioral data (self-care behavior, physical activity, dietary assessment, smoking, alcohol consumption, and sleep quality), and psycho-social data (depression and DRD). Poor glycemic control was defined as not achieving the target HbA1c based on the 2021 American Diabetes Association (ADA) Guideline. Multivariable logistic regression was used to explore the associations between potential factors including DRD, and poor glycemic control.

### Results

The prevalence of poor glycemic control in patients with type 2 diabetes was 29.1%. Our analysis revealed that age under 65 years old (OR 6.40, 95% CI 2.07–19.77, p = 0.001), obesity (BMI ≥ 25 kg/m$^2$) (OR 2.96, 95% CI 1.05–8.39, p = 0.041), and DRD (OR 14.20, 95% CI 3.76–53.64, p<0.001) were significantly associated with poor glycemic control.

**Data Availability Statement:** All relevant data are within the paper and its Supporting Information files.

**Funding:** This study was supported by Faculty of Medicine, Chiang Mai University grant number 035/2566. https://w1.med.cmu.ac.th/research/Homepage/Default.html#. The funders had no role in study design, data collection and analysis, decision to publish, or preparation of the manuscript.

**Competing interests:** The authors have declared that no competing interests exist.

Three dimensions of DRD were associated with poor glycemic control, including emotional distress (OR 4.23, 95% CI 1.51–11.85, p = 0.006), regimen-related distress (OR 6.00, 95% CI 1.88–19.18, p = 0.003), and interpersonal distress (OR 5.25, 95% CI 1.39–20.02, p = 0.015).

## Conclusion and recommendation

Age, obesity, and DRD are associated with poor glycemic control. A holistic approach that includes addressing DRD is crucial for improving glycemic outcomes in patients with type 2 diabetes. Further studies in broader populations using a cohort design are recommended.

## Introduction

Type 2 diabetes mellitus (DM) is a major global health concern with a high prevalence of 462 million cases in 2017 [1], and is expected to rise to 643 million by 2030 [2]. Uncontrolled diabetes can cause many serious complications including retinopathy, nephropathy, neuropathy, and cardiovascular diseases [3], which impact the quality of life, increasing mortality and raising healthcare expenditures [4]. Effective glycemic control is important to prevent diabetic complications. To ensure this, guidelines have been continuously updated to focus on maintaining optimal glycemic levels. However, poor glycemic control remains widespread, i.e., the prevalence of uncontrolled diabetes was 47.3% in Brazil [5], 50.2% in Japan [6], and 76.9% in Saudi Arabia [7]. Thailand ranks fifth in the number of DM cases among Western Pacific countries [2], with 64.4% of DM cases being poorly controlled [8]. In Northern Thailand, the prevalence of poor glycemic control among DM cases was 54.8% [9].

Achieving a glycemic goal is difficult due to numerous factors, which can be divided into demographic factors, clinical factors, behavioral factors, and psycho-social factors. For demographic aspect, factors associated with poor glycemic control include age [8, 10, 11] and sex [8]. For clinical aspect, there are many factors associated with poor glycemic control, including obesity [12, 13], duration of diabetes [7, 8], and diabetes treatment modalities [5, 11]. In the behavioral aspect, factors associated with poor glycemic control include self-care behaviors [12], energy intake [14], physical activities [5, 15, 16], sleeping quality [17], and smoking [7]. In the psycho-social aspect, depression is a common psychological issue associated with poor glycemic control [18]. However, evaluation for depression does not cover the social aspect, which is also important for diabetes treatment outcomes. Therefore, diabetes-related distress (DRD) is one of the important concerns [19–21].

DRD refers to a broader affective experience than depression, as it covers both psychological and social aspects. DRD implies worries, concerns, and fears related to the burden of living with diabetes, such as feeling overwhelmed by diabetes management tasks or experiencing negative emotions related to the condition [22]. The prevalence of DRD was 29.4% in Vietnam [23], 37.0% in Saudi Arabia [19], and 39% in Canada [24]. Furthermore, DRD is associated with poor self-management [19, 21] and poor glycemic outcomes [19–21]. Despite the importance of DRD in diabetes management, it is often overlooked in clinical settings. Therefore, attending to psycho-social concerns may be the key to achieve the glycemic goal. DRD screening may be an important step in dealing with psycho-social problems in diabetic patients.

While literature has extensively studied demographic, clinical, and behavioral factors affecting glycemic control in patients with type 2 diabetes [5, 7, 8, 10–17], there is a knowledge gap concerning psycho-social factors in Thailand. Particularly, the impact of DRD, encompassing

both psychological and social aspects, has not been adequately investigated. Therefore, this study aims to investigate: 1) the prevalence and associated factors of poor glycemic control type 2 diabetes, and 2) the prevalence of DRD and the association of its subcomponents and poor glycemic control type 2 diabetes in Maharaj Nakorn Chiang Mai Hospital, Chiang Mai, Thailand.

## Materials and methods

### Study design, setting, and period

A cross-sectional study was conducted from 1st December 2021 to 18th March 2022 at the Outpatient Clinic of the Family Medicine Department, Maharaj Nakorn Chiang Mai Hospital. The hospital is affiliated with the Faculty of Medicine, Chiang Mai University, and functions as both a training center and a tertiary medical care facility for patients in Chiang Mai and 16 other Northern Thai provinces. The outpatient clinic of the Family Medicine department serves as a primary care facility for non-urgent medical conditions. The clinic registers and treats about 9,000 patients each year, with approximately 1,900 cases being type 2 diabetes patients.

### Population

The study population consisted of type 2 diabetic patients registered in the family medicine outpatient clinic at our institute. The patients received treatment based on routine guidelines, which include both medication with lifestyle modification and lifestyle modification only. Inclusion criteria were as follows: 1) known cases of type 2 diabetic patients with scheduled appointments and glycated hemoglobin (HbA1c) tests during the study period, 2) aged at least 18 years old, and 3) able to answer the questionnaire and agree to participate in the study. The exclusion criterion was a recent diagnosis of type 2 diabetes within the past 6 months.

### Sampling and sample size estimation

Patients were recruited through consecutive sampling. The sample size was calculated using the formula for estimating a proportion in an infinite population [25]. According to relevant literature, we anticipated that about 54.8% of type 2 diabetic patients would have poor glycemic control [9]. We defined a margin of error (d) of 0.10 and a significance level (alpha) of 0.05, with Z(0.975) = 1.96. The calculated sample size was 96 patients. Additionally, an additional 20% was added to account for incomplete responses or missing data.

$$n = \frac{z_{1-\frac{\alpha}{2}}^2 p(1-p)}{d^2}$$

### Research instrument and measures

Data, including age, sex, occupation, personal monthly income, education, marital status, health insurance, smoking, alcohol consumption, comorbidities, and duration of being type 2 diabetes, were collected via a questionnaire.

Cognitive screening was performed using the Thai version of the brief cognitive screening test (Mini-Cog) (Cronbach's $\alpha$ = 0.80) [26, 27], with a score of $\leq$ 3 indicating cognitive impairment.

Self-care behavior data were assessed using a questionnaire developed by Siangdung et al (Cronbach's $\alpha$ = 0.70) [28]. This questionnaire consists of 20 questions covering 5 behaviors, including eating, medical adherence, exercise, dealing with stress, and continuing of treatment. Each question is rated on a scale of 1 (not performing), 2 (occasionally performing), and 3

(regularly performing). The mean score for each behavior was categorized into three self-care groups: low (1.00–1.66), medium (1.67–2.33) and high (2.34–3.00).

Physical activity data were collected using the Thai version of the Global Physical Activity Questionnaires (GPAQ) (Cronbach's α = 0.82) [29, 30], which recorded physical activity data and then calculated the metabolic equivalent (MET). The World Health Organization (WHO) recommends that adequate physical activity includes at least 150 minutes of moderate-intensity or 75 minutes of vigorous-intensity physical activity per week, or at least 600 MET-minutes per week [30].

Dietary assessment was collected using a 24-hour dietary recall. The research assistant nurse instructed the patients to provide information about the type, name, and amount of food and drink consumed in the previous 24 hours. Total calorie and carbohydrate intake were analyzed by a dietitian, while other data were kept blind to prevent bias.

Sleep quality in the past 1 month was evaluated using the Thai version of the Pittsburgh Sleep Quality Index (PSQI) (Cronbach's α = 0.84) [31]. This questionnaire consists of 19 questions, with the total score ranging from 0 to 21. A score of ≥5 indicates poor sleep quality.

Depression screening was assessed using 9-Questions Depression Rating Scale (9Q) [32]. This screening tool consists of 9 questions about depressive symptoms experienced within the last 2 weeks. A score of ≥7 indicates a positive result for depression on the 9Q.

DRD was assessed using the Thai version of the Diabetes Distress Scale questionnaire (DDS-17) (Cronbach's α = 0.95) [33, 34]. This questionnaire consists of 17 questions in 4 different dimensions (emotional distress, regimen-related distress, physician-related distress, and diabetes-related interpersonal distress). Each question is rated on a scale from 1 (no problems) to 6 (serious problems). Each dimension was interpreted based on its mean score, resulting in two groups: no distress ($< 2$) and moderate to high distress ($\geq 2$).

Moreover, data including weight, height, waist circumference, blood pressure, fasting blood sugar (FBS), glycated hemoglobin (HbA1c), and diabetes treatment modalities were obtained from the electronic medical record. For FBS and HbA1c, whole blood specimens were collected by licensed medical technicians on the same day as the data collection. Patients fasted (nothing per oral, NPO) for 12 hours prior to blood collection. All specimens were properly stored and analyzed by our central laboratory on the same day. The results were then recorded in the electronic medical record. Body mass index (BMI) was calculated by dividing a person's weight in kilograms by their height in meters squared. In this study, a BMI of $\geq 25$ kg/m2 was considered obese [35].

## Data collection

Data collection was carried out within a single session at the outpatient clinic. All type 2 diabetic patients who visited the clinic and met the inclusion criteria were voluntarily invited to participate by the outpatient clinic's nurse. Participants who were willing to participate in the study were directed to a registered nurse who served as a research data collector and was trained to conduct interviews. The registered nurse provided information about the data collection process and obtained written informed consent from the patients.

Subsequently, demographic data were collected through a self-administered paper-based questionnaire, which included information such as age, sex, occupation, personal monthly income, education, marital status, and health insurance. The information that patients had completed was then submitted to the same registered nurse for a thorough check to ensure completeness.

The same registered nurse conducted questionnaire interviews and performed assessments using standard tools, including comorbidities, duration of type 2 diabetes, smoking, alcohol

consumption, cognitive screening (Mini-Cog), self-care behaviors (self-care behavior questionnaire), physical activity (GPAQ), dietary assessments (24-hour dietary recall), sleep quality (PSQI), depression screening (9Q), and DRD assessment (DDS-17).

Our registered nurse received training in the use of these questionnaires and assessment tools before starting data collection. In Thailand, registered nurses are qualified to conduct screening tests, such as the self-care behavior questionnaire [28], Mini-Cog [27], GPAQ [29], PSQI [31], 9Q [32], and DDS-17 [33], under the supervision of a family physician. If any abnormalities were detected during the data collection, the assistant nurse would promptly notify the attending family physicians. Otherwise, patients would continue their routine examinations and treatment. In this study, none of the attending family physicians at the outpatient clinic were involved in the data collection process.

Afterward, a research data collection assistant, who was not involved in the collection of questionnaire and patients' treatment, retrieved data including weight, height, waist circumference, blood pressure, FBS, HbA1c, and diabetes treatment modalities from the electronic medical records of that day. Another research assistant was assigned to double-check the data for completeness, consistency, reliability, and to prevent transcription errors. Subsequently, only dietary assessment data (24-hour dietary recall) were sent to a dietitian for evaluating total calorie and carbohydrate intake, and these were later returned to the main investigator. Only the investigator team had access to information that could identify individual participants during data collection.

## Outcome of interest

Glycemic control was determined according to The American Diabetes Association (ADA) Guideline 2021 [36]. The target HbA1c was determined based on age and comorbidities, including: 1) HbA1c < 7.0% (age ≤ 65 years old), 2) HbA1c < 7.5% (age > 65 years old, independent, without cognitive impairment), 3) HbA1c < 8.0% (age > 65 years old with ≥ 3 comorbidities (e.g., hypertension, arthritis, cancer, heart failure, falls, CKD stage ≥ 3, myocardial infarction, stroke), frailty, dementia, dependent). Patients who failed to meet the HbA1c target were classified as having poor glycemic control, whereas those who achieved it were classified as having good glycemic control.

## Statistical analysis

The data were analyzed by using Stata 16.0 (StataCorp, College Station, Texas, USA). Descriptive statistics were expressed as frequencies and percentage for categorical data, mean ± SD or median (IQR) for numerical data depending on the underlying distribution. Independent t-test, Mann-Whitney U test and Fisher's exact test were used to evaluate the significant differences of variables between good and poor glycemic control groups. Factors with statistically significant results from univariable analysis (p-value < 0.05) were further explored using multivariable logistic regression analysis. Additionally, multivariable logistic regression analysis was also used to explore the association between DRD subcomponents and poor glycemic control while adjusting for sex and other factors previously identified as significant in the univariable analysis.

## Ethical considerations

Ethical approval was obtained from the Research Ethics Committee, Faculty of Medicine, Chiang Mai University, Thailand (FAM-2564-08500). Written informed consent was obtained from the study participants after they were informed of the study's purpose.

## Results

A total of 160 diabetes patients visited Outpatient Clinic of the Family Medicine Department, Maharaj Nakorn Chiang Mai Hospital during the study period. Among them, 33 patients were excluded: 4 patients were newly diagnosed with type 2 diabetes, and 29 patients did not volunteer to participate in the study. Therefore, 127 patients voluntarily participated in this study.

The mean age was 66.2 ± 7.3 years, ranging from 44 to 83 years old, with female predominance (52%). Most of them were married (73.2%). Median duration of being type 2 diabetes was 8 years (IQR 3,11 years). Among the participants, 88.2% were receiving medical treatment with lifestyle modification, while 11.8% were on lifestyle modification only. Cognitive impairment, as assessed by the Mini-Cog test, was not present in most participants (94.5%). The mean HbA1c was 6.9 ± 0.8% and the prevalence of poor glycemic control type 2 diabetes was 29.1% (31/127). Table 1 presents the comparison of demographic and clinical factors between poor glycemic control and good glycemic control type 2 diabetes. The poor glycemic control group had higher prevalence of age under 65 years old (p-value = 0.014), obesity (p-value = 0.012), using sulfonylurea (p-value = 0.029), and using two or more types of diabetes medication (p-value = 0.012) (**Table 1).**

Concerning behavioral and psycho-social factors, most patients had high level of self-care behavior (84.3%). Regarding medical adherence, a subcomponent of self-care behavior, there were 112 patients receiving medical treatment, and most of them (99.1%) exhibited a high level of medical adherence (110/112). Notably, none of the patients used long-term corticosteroids. Moreover, 22.1% of patients had inadequate physical activity, and 37.7% experienced poor sleep quality. The prevalence of depression and DRD among patients in this study was 6.3% (8/127) and 19.7% (25/127), respectively. Table 2 presents the comparison of behavioral and psycho-social factors between poor glycemic control and good glycemic control type 2 diabetes. In terms of behavioral factors, there were no significant differences in self-care behavior, medical adherence, physical activity, dietary intake, smoking, and alcohol consumption between the two groups. Interestingly, psycho-social factors were particularly remarkable. The poor glycemic control group had higher prevalence of being diabetic patient with distress (p-value<0.001), depression (p-value = 0.046), and poor sleep quality (p-value = 0.008) (**Table 2**).

Table 3 presents multivariable logistic regression analysis of association between poor glycemic control type 2 diabetes and potential related factors. Factors significantly associated with poor glycemic control type 2 diabetes were age under 65 years old (p-value = 0.001), obesity (p-value = 0.041), and being diabetic patient with distress (p-value<0.001) (**Table 3**).

Table 4 presents the comparison of DDS-17 dimensions between poor glycemic control and good glycemic control type 2 diabetes. The prevalence of emotional distress was 26.0%, regimen-related distress was 22.1%, physician-related distress was 3.2%, and diabetes-related interpersonal distress was 12.6%. Moreover, the group with poor glycemic control had higher prevalence of emotional distress (p-value<0.001), regimen-related distress (p-value<0.001) and diabetes-related interpersonal distress (p-value = 0.001) (**Table 4**).

Table 5 presented multivariable logistic regression analysis of association between poor glycemic control type 2 diabetes and DDS-17 dimensions. Emotional distress (p-value = 0.006), regimen-related distress (p-value = 0.003), and diabetes-related interpersonal distress (p-value = 0.015) were associated with uncontrolled type 2 diabetes after adjusted for age, sex, obesity, using sulfonylureas, using two or more types of diabetes medication, depression, and poor sleep quality (**Table 5**).

**Table 1. Comparison of demographic and clinical factors between poor glycemic control and good glycemic control type 2 diabetes (n = 127).**

| Factor | DM treatment outcome | | | | p-value |
|---|---|---|---|---|---|
| | Uncontrolled (n = 37) | | Controlled (n = 90) | | |
| | n | % | n | % | |
| **Demographic factors** | | | | | |
| Age | | | | | |
| Age < 65 years old | 19 | 51.4 | 25 | 27.8 | |
| Age ≥ 65 years old (elderly) | 18 | 48.6 | 65 | 72.2 | 0.014 [c] |
| Sex | | | | | |
| Male | 16 | 43.2 | 45 | 50.0 | |
| Female | 21 | 56.8 | 45 | 50.0 | 0.560 [c] |
| Occupation | | | | | |
| In employment/working | 16 | 43.2 | 32 | 35.6 | |
| Unemployed/not working | 21 | 56.8 | 58 | 64.4 | 0.428 [c] |
| Personal monthly income (Thai baht) (median (IQR)) | 20000 | (10000,32000) | 16384 | (10000,30000) | 0.280 [b] |
| Education level | | | | | |
| Lower than bachelor's degree | 20 | 54.1 | 50 | 55.6 | |
| Bachelor's degree or higher | 17 | 46.0 | 40 | 44.4 | 1.000 [c] |
| Marital status | | | | | |
| Married | 26 | 70.3 | 67 | 74.4 | |
| Single, separated, divorced, widow | 11 | 29.7 | 23 | 25.6 | 0.663 [c] |
| Health Insurance | | | | | |
| Government | 36 | 97.3 | 85 | 94.4 | |
| Non-government | 1 | 2.7 | 5 | 5.6 | 0.671 [c] |
| **Clinical factors** | | | | | |
| Comorbidities | | | | | |
| > 1 other underlying diseases | 30 | 81.1 | 81 | 90.0 | 0.237 [c] |
| Cognitive screening by Mini-Cog | | | | | |
| No cognitive impairment (Mini-Cog > 3) | 34 | 91.9 | 86 | 95.6 | |
| Cognitive impairment (Mini-Cog ≤ 3) | 3 | 8.1 | 4 | 4.4 | 0.415 [c] |
| Body weight (kg) (mean ± SD) | 65.6 | ± 11.5 | 63.5 | ± 12.0 | 0.378 [a] |
| Height (cm) (mean ± SD) | 157.8 | ± 8.3 | 159.4 | ± 8.5 | 0.329 [a] |
| BMI (kg/m$^2$) (mean ± SD) | 26.4 | ± 4.3 | 24.9 | ± 3.6 | 0.049 [a] |
| Obesity (BMI ≥ 25 kg/m2) | 24 | 64.9 | 36 | 40.0 | 0.012 [c] |
| Waist circumference (cm) (mean ± SD) | 88.4 | ± 9.5 | 87.4 | ± 10.0 | 0.621 [a] |
| Systolic blood pressure(mmHg) (mean ± SD) | 140.5 | ± 10.7 | 136.5 | ± 13.1 | 0.098 [a] |
| Diastolic blood pressure(mmHg)(mean ± SD) | 79.9 | ± 8.8 | 78.4 | ± 9.4 | 0.422 [a] |
| Duration of DM (years) (median (IQR)) | 9 | (4,10) | 8 | (3,11) | 0.845 [b] |
| Diabetes treatment | | | | | |
| Lifestyle modification only | 4 | 10.8 | 11 | 12.2 | |
| Medical treatment with lifestyle modification | 33 | 89.2 | 79 | 87.8 | 1.000 [c] |
| Type of medication treatment (single or combination regimen) | | | | | |
| Biguanide | 32 | 86.5 | 76 | 84.4 | 1.000 [c] |
| Sulfonylureas | 15 | 40.5 | 19 | 21.1 | 0.029 [c] |
| Thiazolidinedione | 2 | 5.4 | 3 | 3.3 | 0.628 [c] |
| DPP-4 inhibitor | 15 | 40.5 | 33 | 36.7 | 0.692 [c] |
| SGLT-2 inhibitor | 2 | 5.4 | 2 | 2.2 | 0.579 [c] |
| Insulin | 0 | 0 | 1 | 1.1 | 1.000 [c] |
| Using ≥ 2 types of DM medication | 24 | 64.9 | 36 | 40.0 | 0.012 [c] |

*(Continued)*

**Table 1.** (Continued)

| Factor | DM treatment outcome | | | | p-value |
|---|---|---|---|---|---|
| | Uncontrolled (n = 37) | | Controlled (n = 90) | | |
| | n | % | n | % | |
| FBS (mg/dL) (mean ± SD) | 155.8 | ± 35.0 | 122.9 | ± 16.2 | < 0.001 [a] |
| HbA1c (%) (mean ± SD) | 7.8 | ±0.6 | 6.5 | ±0.5 | <0.001 [a] |

[a] Independent t-test

[b] Mann-Whitney U test

[c] Fisher's exact test

**Table 2. Comparison of behaviors and psycho-social factors between poor glycemic control and good glycemic control type 2 diabetes (n = 127).**

| Factor | DM treatment outcome | | | | p-value |
|---|---|---|---|---|---|
| | Uncontrolled (n = 37) | | Controlled (n = 90) | | |
| | n | % | n | % | |
| **Behavioral factors** | | | | | |
| Current smoking | 4 | 10.8 | 2 | 2.2 | 0.059 [c] |
| Current alcohol drinking | 13 | 35.1 | 20 | 22.2 | 0.181 [c] |
| Frequency of alcohol drinking (n = 33) | | | | | |
| ≤ 1 time per month | 7 | 53.8 | 8 | 40.0 | |
| 2–4 times per month | 2 | 15.4 | 4 | 20.0 | |
| >2 times per week | 4 | 30.8 | 8 | 40.0 | 0.810 [c] |
| Amount of alcohol consumed (Standard drinks per time) (median (IQR)) | 3.2 | (1.28,5) | 2 | (1,4.5) | 0.401 [b] |
| Self-care behavior (Total Health Behavior score) (mean ± SD) | 2.5 | ± 0.2 | 2.6 | ± 0.2 | 0.221 [a] |
| High (score 2.34–3.00) | 31 | 83.8 | 76 | 84.4 | |
| Medium (score 1.67–2.33) | 6 | 16.2 | 14 | 15.6 | |
| Low (score 1.00–1.66) | 0 | 0 | 0 | 0 | 1.000 [c] |
| Medical adherence score (n = 112) (mean ± SD) | 2.6 | ± 0.2 | 2.7 | ± 0.2 | 0.152 [a] |
| High (score 2.34–3.00) | 32 | 97.0 | 78 | 98.7 | |
| Medium (score 1.67–2.33) | 1 | 3.0 | 1 | 1.3 | |
| Low (score 1.00–1.66) | 0 | 0 | 0 | 0 | 0.504 [c] |
| Physical activity (MET score) (median (IQR)) | 880 | (560,2860) | 1120 | (720,2120) | 0.538 [b] |
| Inadequate physical activity | 11 | 29.7 | 17 | 18.9 | 0.238 [c] |
| 24 hr. Dietary Recall | | | | | |
| Carbohydrate (kcal/day) (mean ± SD) | 829.9 | ±347.4 | 851.9 | ±332.7 | 0.738 [a] |
| Total calory (kcal/day) (mean ± SD) | 1598.6 | ±463.9 | 1657.3 | ±510.0 | 0.546 [a] |
| Sleep quality by PSQI score | | | | | |
| Poor sleep quality (PSQI >5) | 21 | 56.8 | 27 | 30.0 | 0.008 [c] |
| **Psycho-social factors** | | | | | |
| Screening Depression by 9Q | | | | | |
| Positive (9Q ≥ 7) | 5 | 13.5 | 3 | 3.3 | 0.046 [c] |
| DRD by DDS-17 score | | | | | |
| No distress (score <2) | 18 | 48.7 | 84 | 93.3 | |
| With distress (score ≥ 2) | 19 | 51.4 | 6 | 6.7 | < 0.001 [c] |

[a] Independent t-test

[b] Mann-Whitney U test

[c] Fisher's exact test

**Table 3. Multivariable logistic regression analysis of association between poor glycemic control type 2 diabetes and potential related factors (n = 127).**

| Factor | Odd Ratio | 95% CI | p-value |
|---|---|---|---|
| Age | | | |
| Age $\geq$ 65 years old (elderly) | Reference | | |
| Age < 65 years old | 6.40 | 2.07–19.77 | 0.001 |
| Sex | | | |
| Male | Reference | | |
| Female | 1.66 | 0.58–4.77 | 0.343 |
| BMI | | | |
| Not obesity (BMI < 25) | Reference | | |
| Obesity (BMI $\geq$ 25) | 2.96 | 1.05–8.39 | 0.041 |
| Using sulfonylureas | | | |
| Not using sulfonylureas | Reference | | |
| Using sulfonylureas | 1.91 | 0.52–7.09 | 0.331 |
| Diabetes medication | | | |
| Lifestyle modification or using 1 type of DM medication | Reference | | |
| Using two or more types of diabetes medication | 1.89 | 0.55–6.47 | 0.313 |
| DRD by DDS-17 score | | | |
| No distress (score < 2) | Reference | | |
| With distress (score $\geq$ 2) | 14.20 | 3.76–53.64 | <0.001 |
| Screening depression (9Q) | | | |
| No depression (9Q < 7) | Reference | | |
| Depression (9Q $\geq$ 7) | 2.88 | 0.31–26.33 | 0.349 |
| Sleep quality by PSQI score | | | |
| Normal sleep quality (PSQI $\leq$ 5) | Reference | | |
| Poor sleep quality (PSQI > 5) | 2.01 | 0.66–6.14 | 0.219 |

**Table 4. Comparison of DDS-17 dimensions between poor glycemic control and good glycemic control type 2 diabetes (n = 127).**

| Factor | DM treatment outcome | | | | p-value |
|---|---|---|---|---|---|
| | Uncontrolled (n = 37) | | Controlled (n = 90) | | |
| | n | % | n | % | |
| Emotional distress | | | | | |
| No distress (score <2) | 18 | 48.7 | 76 | 84.4 | |
| With distress (score $\geq$ 2) | 19 | 51.3 | 14 | 15.6 | <0.001 [c] |
| Regimen-related distress | | | | | |
| No distress (score <2) | 19 | 51.3 | 80 | 88.9 | |
| With distress (score $\geq$ 2) | 18 | 48.7 | 10 | 11.1 | <0.001 [c] |
| Physician-related distress | | | | | |
| No distress (score <2) | 35 | 94.6 | 88 | 97.8 | |
| With distress (score $\geq$ 2) | 2 | 5.4 | 2 | 2.2 | 0.579 [c] |
| Diabetes-related interpersonal distress | | | | | |
| No distress (score <2) | 26 | 70.3 | 85 | 94.4 | |
| With distress (score $\geq$ 2) | 11 | 29.7 | 5 | 5.6 | 0.001 [c] |

[c] Fisher's exact test

**Table 5. Multivariable logistic regression analysis of association between poor glycemic control type 2 diabetes and DDS-17 dimensions (n = 127).**

| Factor | Odd Ratio | 95% CI | p-value |
|---|---|---|---|
| Emotional distress | | | |
| No distress (score <2) | Reference | | |
| With distress (score ≥ 2) | 4.23 | 1.51–11.85 | 0.006 |
| Regimen-related distress | | | |
| No distress (score <2) | Reference | | |
| With distress (score ≥ 2) | 6.00 | 1.88–19.18 | 0.003 |
| Physician-related distress | | | |
| No distress (score <2) | Reference | | |
| With distress (score ≥ 2) | 1.22 | 0.11–13.23 | 0.869 |
| Diabetes-related interpersonal distress | | | |
| No distress (score <2) | Reference | | |
| With distress (score ≥ 2) | 5.25 | 1.38–20.02 | 0.015 |

Adjust for age, sex, obesity, using sulfonylureas, using two or more types of diabetes medication, depression, and poor sleep quality.

## Discussion

The purpose of this study was to investigate the prevalence of poor glycemic control in patients with type 2 diabetes and its association with clinical factors and diabetes-related distress in a primary care setting. The prevalence of poorly controlled type 2 diabetes in our study was 29.1%, which was lower than in previous studies: 47.3% in Brazil [5], 50.2% in Japan [6], and 84.3% in Uganda [10]. The difference in prevalence might be attributed to variations in HbA1c target goals. Some countries adopt an HbA1c goal of less than 7% [6, 10], while others follow the HbA1c goal outlined in the ADA guideline for 2021 [5], which allows for greater flexibility, especially for older patients. The sociodemographic characteristics of patients and the university affiliation may have influenced the lower prevalence of poor glycemic control [5].

From our analysis, we have identified three factors independently associated with poor glycemic control. The first factor was being under 65 years old, which was consistent with findings from previous studies [8–11]. This association can be explained by the fact that younger to middle-aged patients may be more occupied with work and more likely to miss doctor's appointments and disregard self-care behaviors, resulting in poor glycemic control [8–10]. Conversely, elderly patients might be more motivated to manage their diabetic conditions [8]. Additionally, this may be due to the flexible HbA1c target range for the elderly in the ADA guideline for 2021 [36]. The second factor, obesity was also supported by previous studies [12, 13] that patients with greater BMI, who were expected to have low physical activity levels, were more likely to experience poor glycemic control compared to those with a normal BMI [12]. Furthermore, evidence supports the hypothesis that obesity was associated with insulin resistance and beta-cell dysfunction, leading to hyperglycemia [37]. Therefore, addressing obesity through weight reduction interventions could potentially improve glycemic control and overall health status in individuals with poorly controlled type 2 diabetes [38].

The last factor was DRD. The prevalence of diabetic patients with DRD in this study was 19.7%, which closely aligns with figures reported in previous studies, such as 22.6% in Greece [39], 25% in Saudi Arabia [40], and 29.4% in Vietnam [23]. However, another study conducted in Canada showed a higher prevalence of DRD, at 39% [24]. This result indicates that the prevalence of DRD varies among different countries and service settings. The association of DRD

with poor glycemic control was in conformity with the results of previous studies conducted in Japan [20], Saudi Arabia [19], and the United States [41]. Previous studies have found that DRD, as opposed to depression, was associated with higher HbA1c levels [20, 21]. DRD was also associated with a low level of autonomy support [41], poor treatment adherence [19], and poor glycemic control [19, 41]. Furthermore, there is evidence supporting the idea that DRD could lead to dysregulation of stress hormones and subsequent hyperglycemia [39].

Regarding the subcomponents analysis of DRD, it was found that emotional distress and regimen-related distress were common among type 2 diabetic patients in this study. The prevalence of emotional distress was 26.0%, regimen-related distress was 22.1%, physician-related distress was 3.2%, and diabetes-related interpersonal distress was 12.6%. Our analysis found a significant association between emotional distress, regimen-related distress, and diabetes-related interpersonal distress with poor glycemic control type 2 diabetes, which was consistent with previous studies [24, 39, 42]. Studies have found an association between emotional distress and poor glycemic control, which could be attributed to the psychological stress of managing their condition [24, 39, 42]. Furthermore, diabetes-related interpersonal distress has also been found to be associated with poor glycemic control [39], possibly due to the challenges of maintaining social relationships while coping with the impact of diabetes on daily life. Additionally, regimen-related distress has been linked to poor medication adherence [43] and higher HbA1c levels [24, 39], which may be related to difficulties in adhering to medication and self-care management. Moreover, evidence supports that a lower level of DRD score is positively associated with better glycemic control [23]. Therefore, the DDS-17 could be used to assess patients' specific concerns and assist in personalized diabetes management plans. Additionally, studies have revealed that interventions such as peer support and brief group cognitive-behavioral therapy can effectively reduce DRD and improve glycemic control in poorly controlled type 2 diabetic patients with DRD [44, 45]. Consequently, appropriate DRD detection and management through screening and the implementation of interventions, such as cognitive-behavioral therapy or supervision, could enhance self-care and improve treatment outcomes for diabetic patients.

In addition to the factors previously discussed, previous studies have also identified other factors associated with poor glycemic control, such as medical adherence [46], depression [18], and poor sleep quality [17]. However, these findings were inconsistent with the results of our study. The high adherence among most of the included patients may have limited the significance of medical adherence in our study. Additionally, depression and sleep quality showed only potential but not significant associations with poor glycemic control, possibly due to the small sample size. Studies with larger sample sizes could provide more robust findings.

The main strength of the study is the comprehensive evaluation of potential factors associated with poor glycemic control in type 2 diabetes across multiple dimensions, including demographic, clinical, behavioral, and psychosocial factors. This approach provides a more holistic view of the factors associated with poor glycemic control. Another strength of the study is the use of standardized tools for data collection and analysis, which enhanced the accuracy and reliability of the results. The expertise of a dietitian also improved the quality of the results, particularly in dietary assessment. Moreover, the utilization of two trained assistants to collect data separately in their assigned areas minimized inconsistencies in data collection. However, this study has some limitations. Firstly, its cross-sectional design does not allow for the confirmation of a causal relationship between factors potentially associated with poor glycemic control in type 2 diabetes. A cohort design, which follows a population at risk over time, may be better suited for causal inference. Secondly, there may have been some recall bias due to the high proportion of elderly patients. However, this bias is likely minimal because most of the patients had no cognitive impairment. Thirdly, as we did not perform an official

sample size estimation for conducting the multivariable analysis, some important factors might not have reached statistical significance. The interpretation of our results should focus on the direction and effect size rather than solely on statistical significance, and further research with a larger study size is encouraged to explore relevant factors and validate our findings more comprehensively. Finally, the study population was limited to patients at our family medicine clinic. This limitation may impact the generalizability of the findings. To enhance the generalizability of the results, future studies should include a more diverse population.

## Conclusion

In conclusion, although the prevalence of poor glycemic control in type 2 diabetes was relatively low in this study, it remains a significant issue deserving attention. We identified factors linked to poor glycemic control, including age under 65, obesity, and DRD. Additionally, three dimensions of DRD (emotional distress, regimen-related distress, and diabetes-related interpersonal distress) were associated with poor glycemic control. These findings emphasize the need for a holistic approach to diabetes management, encompassing demographic, clinical, behavioral, and psychosocial factors in patient care.

## Supporting information

**S1 File. Study flow.**
(DOCX)

## Acknowledgments

The authors thank the Family Medicine Department, Maharaj Nakorn Chiang Mai Hospital for their support, the OPD family medicine staff for their help in patient recruitment, and all participants for providing informative data and giving permission for its publication.

## Author Contributions

**Conceptualization:** Achiraya Ruangchaisiwawet, Phichayut Phinyo, Nalinee Yingchankul.

**Data curation:** Achiraya Ruangchaisiwawet, Nalinee Yingchankul.

**Formal analysis:** Achiraya Ruangchaisiwawet, Krittai Tanasombatkul, Phichayut Phinyo, Nalinee Yingchankul.

**Funding acquisition:** Nalinee Yingchankul.

**Investigation:** Achiraya Ruangchaisiwawet, Narumit Bankhum, Nalinee Yingchankul.

**Methodology:** Achiraya Ruangchaisiwawet, Phichayut Phinyo, Nalinee Yingchankul.

**Project administration:** Nalinee Yingchankul.

**Resources:** Nalinee Yingchankul.

**Software:** Achiraya Ruangchaisiwawet, Krittai Tanasombatkul, Nalinee Yingchankul.

**Supervision:** Phichayut Phinyo, Nalinee Yingchankul.

**Validation:** Nalinee Yingchankul.

**Visualization:** Nalinee Yingchankul.

**Writing – original draft:** Achiraya Ruangchaisiwawet, Nalinee Yingchankul.

**Writing – review & editing:** Achiraya Ruangchaisiwawet, Phichayut Phinyo, Nalinee
Yingchankul.

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
