## [Decision Letter · Decision Letter 0]

18 Jun 2023

PONE-D-23-07649Physical, behavioral, and psychosocial factors associated with uncontrolled type 2 diabetes: a Northern Thai cross-sectional study.PLOS ONE

Dear Dr. yingchankul,

Thank you for submitting your manuscript to PLOS ONE. After careful consideration, we feel that it has merit but does not fully meet PLOS ONE’s publication criteria as it currently stands. Therefore, we invite you to submit a revised version of the manuscript that addresses the points raised during the review process. Please note the comments made by the 2 reviewers below and in the attached file and make the necessary amendments. 

We look forward to receiving your revised manuscript.

Kind regards,

Shairyzah Ahmad Hisham, PhD.

Academic Editor

PLOS ONE

2.Thank you for stating the following in your Competing Interests section: 

“The authors declare no conflict of interest.”

Reviewers' comments:

**Comments to the Author**

1. Is the manuscript technically sound, and do the data support the conclusions?

Reviewer #1: No

Reviewer #2: Partly

2. Has the statistical analysis been performed appropriately and rigorously? 

Reviewer #1: No

Reviewer #2: I Don't Know

3. Have the authors made all data underlying the findings in their manuscript fully available?

Reviewer #1: Yes

Reviewer #2: Yes

4. Is the manuscript presented in an intelligible fashion and written in standard English?

Reviewer #1: Yes

Reviewer #2: Yes

5. Review Comments to the Author

Reviewer #1: Dear author,

Although I think this is a good study that could offer new information and potentially improve diabetic care among patients, there were a few major parts of the manuscript that require revision.

Please kindly refer to the attached document. The document is the submitted manuscript containing track changes and suggestions.

Reviewer #2: Dear authors, congratulations on completing the research and producing an illustrative manuscript!

My comments only serve to improve the clarity of reading and transparency of scientific reporting.

Title

The use of the word “Physical factor” in this manuscript may be misleading and not according to usual diabetes mellitus literature. Please consider using conventional terminology.

Abstract

Aside from “lack of studies,” please justify the need to study DRD in Thai diabetic population using other reasons.

Please use the signpost method: A→B, B→C, C→D, in order to write a cohesive background. At present, the background consists of 3 disjointed/non-cohesive sentences.

Please re-consider the usage of the words “physical data” and use “sociodemographic, clinical, etc. data” instead. “Diabetic information” can be rephrased as “disease-related information” or simply “clinical data”.

Please re-write the conclusion with better English sentence structure.

Introduction

Line 66 – Please improve sentence to “is of importance” or “is important”

Line 73 – Please justify the use of “physical factors” to denote age, sex, BMI, duration of diabetes and treatment modalities. None of the cited references that were included in the reference list had used the term “physical factors.” Physical factors usually denote physical activity or physical environment. Please go through the literature and use conventional definitions such as demographics, socioeconomics, sociodemographic, clinical, lifestyle-related, disease-related etc..

Line 81 – Please consider removing the word “trending”. Two out of the 3 citations ranged from earlier publications i.e. year 2010 and year 2012, implicating that diabetes-related distress is not a new concern.

Line 82 – The introduction is DRD heavy, with a whole dedicated paragraph to it. It is also one of the objectives and makes up one-third of the results, with its own logistic regression table. It deserves an emphasis whether in the title or in the objective.

Line 97 – Please separate between the prevalence of uncontrolled type-2 diabetes mellitus and prevalence of risk factors for uncontrolled type-2 diabetes mellitus. At present, the objectives are not stated clearly enough.

Line 93 & 94 – Please support your sentence with several citations.

Materials and methods

Line 113 – Please state whether all type 2 diabetic patients are recruited, regardless whether they are receiving pharmacological management or not.

Line 115 – Please clarify the definition of newly diagnosed i.e., 1 month? 1 year? etc. Please omit automatic exclusions such as “unwillingness to answer…” because your inclusion criteria require “willingness to answer.”

Line 119 – Please justify the choice of this sample size formula. Please clarify whether the amount of diabetic outpatients were considered infinite or finite population. Please confirm whether the final sample size was sufficient for running a regression analysis.

Line 120 – Please state (cite) where the reference value of “50.2%” was obtained from and whether it was specific to Thailand/Northern Thailand. If not, please justify the selection of that particular value.

Line 124 – Please state in details whether the complete data collection process was conducted in one session or multiple small sessions. Please state who conducted the data collection, how long did it take, where did it take place. Was each section an individual interview or a patient self-reported questionnaire?

Line 127 – For data extracted from medical records, please state which values were taken. Was data extracted on the same day as patient interview? Was data from upon diagnosis, or the latest value or otherwise? Please state whether extraction was done uniformly across study population. Was the person who the extracted data from medical records also the same person who conducted the interview? Was there any counterchecking to prevent errors in transcribing?

Line 128 – As HbA1c is the most important outcome data in this study, describe in detail how HbA1c level is measured and recorded into the medical file (manual? electronic?)

Line 153 – Depression is a medical diagnosis. Please state whether depression is physician-diagnosed depression or based on obtaining a score in a questionnaire. Was the questionnaire administrator for 9Q trained or qualified to conduct the questionnaire (i.e. must is be a doctor with training in psychiatry)?

Line 187 – Please also state the age range, to better describe the population demographics.

Line 189 – Please state the formula for calculation of prevalence values (uncontrolled DM / DRD/ etc.) in your study.

Table 1 & Table 2

- For occupation, please change “Job / No job” to “In employment/Working/Unemployed/Not working” etc.

- For education level, was the p-value reported twice?

- Please clarify whether Married, living with family is the same as Married (as opposed to Single/not married)

- For diabetes treatment, explain how p-value was derived. Is it a crosstabulation of all treatment or by each treatment type? Please justify whether crosstabulation by each treatment type is appropriate.

- Please denote tests used for deriving p-values. Please confirm the test used for analysing income, duration of DM and physical activity.

Line 215 and 235 – Please consider reporting the regression model significance and fit.

Line 239 – Please state how adjustment was carried out

Discussion

- Please limit stating the results multiple times as they were already stated in the results section in paragraph and also in the tables.

- The discussion was written in a particular format: “The following factors (x,y,z) was associated with uncontrolled type 2 diabetes mellitus, this was consistent with other studies (A,B,C…)”. Although not incorrect, however, the discussion can be improved by using different variations of the format.

- Results were referenced with other supporting studies. However, were there any inconsistent results or opposing studies? Please elaborate on why your results were unique to your community.

- Please discuss the effect of confounders on the results of your study. For example, medication adherence plays an important role in the control of glucose levels, was medication adherence assessed? Similarly, were medication doses optimised and were the patients treated with appropriate medications according to the guidelines i.e. monotherapy, combination OAD, intensive insulin. If these issues are not addressed or controlled for – it can be said that that failure to achieve HbA1c targets may also come from medication under-optimisation and medication non-compliance. How about being on concurrent medications that increases blood sugar levels such as long-term corticosteroids?

Conclusions

- Conclusions were well-written.

General comments

Although English was overall of good level, typos and grammar improvements are needed here and there. Professional proofreading is recommended to elevate manuscript quality.

Majority of the references are recent publications (publish within 10 years).

---

## [Author Response · Author response to Decision Letter 0]

27 Jul 2023

To reviewer 1: Thank you for the opportunity. We have incorporated all reviewers' suggestions into the manuscript revision. Moreover, we have added separate file responses to reviewers 1 and 2.

To reviewer 2: Thank you for the opportunity. We have incorporated all reviewers' suggestions into the manuscript revision. Moreover, we have added separate file responses to reviewers 1 and 2.

---

## [Decision Letter · Decision Letter 1]

21 Aug 2023

PONE-D-23-07649R1Prevalence and associated factors of poor glycemic control type 2 diabetes: a Northern Thai cross-sectional study.PLOS ONE

Dear Dr. yingchankul,

Thank you for submitting your manuscript to PLOS ONE. After careful consideration, we feel that it has merit but further minor revision will help to produce a manuscript of better clarity and standard. Therefore, we invite you to submit a revised version of the manuscript that addresses the points raised during the review process. Please refer to the reviewers' comments at the end of this email.

We look forward to receiving your revised manuscript.

Kind regards,

Shairyzah Ahmad Hisham, PhD.

Academic Editor

PLOS ONE

Journal Requirements:

Reviewer's Responses to Questions

**Comments to the Author**

1. If the authors have adequately addressed your comments raised in a previous round of review and you feel that this manuscript is now acceptable for publication, you may indicate that here to bypass the “Comments to the Author” section, enter your conflict of interest statement in the “Confidential to Editor” section, and submit your "Accept" recommendation.

Reviewer #1: All comments have been addressed

Reviewer #2: Partly

2. Is the manuscript technically sound, and do the data support the conclusions?

Reviewer #1: Yes

Reviewer #2: Partly

3. Has the statistical analysis been performed appropriately and rigorously? 

Reviewer #1: Yes

Reviewer #2: No

4. Have the authors made all data underlying the findings in their manuscript fully available?

Reviewer #1: Yes

Reviewer #2: Yes

5. Is the manuscript presented in an intelligible fashion and written in standard English?

Reviewer #1: Yes

Reviewer #2: No

6. Review Comments to the Author

Reviewer #1: Dear authors,

Thank you for considering all suggestions made in my first reviewed version of the manuscript. I would like to add a few minor suggestions for further improvement of the manuscript. Please refer to below:

1. HbA1C - to change case for 'C'  HbA1c

2. Acronyms - to introduce the acronyms e.g., DRD, ADA etc. once in the manuscript.

Other than that, the R1 version is a more easy-read version and it highlights the amount of work done by the researchers.

Thank you and all the best.

Reviewer #2: Congratulations on the submission of the reviewed manuscript. Thank you for making the required corrections. My comments serve to improve the clarity of reading and transparency of scientific reporting.

Title: Consider adding DRD in the title to reflect the large portion of research and discussion on DRD in the study and manuscript.

Introduction: Sufficiently written to provide background for justification of research

Objectives: Clear

Method:

Line 121: Please change typo in “Estimating and infinite population…” to “Estimating an infinite population…”

Instrument and data collection:

This subsection requires re-writing for clarity and transparency. My concerns are 1) English language, grammar and writing and 2) details – There were various data collection and scales being used. Was/were the data collector(s) trained and eligible to use the various tools/scales? Or was specifically-trained person required i.e. medical doctor etc.?

Line 129 – line 131: Please consider combining into 1 sentence and state who (research assistant) and how data was collected (extraction from database etc.)

Line 132 – State who did the cognitive screening (state the qualification, doctor? Nurse? Bachelor or graduate student?) and whether that person is eligible to do the screening.

Line 139 – Is this “questionnaire” a patient-reported questionnaire or an interviewer-guided questionnaire? Please state in manuscript. Please state whether the person who the extracted data from medical records also the same person who conducted the interview? Was there any counterchecking to prevent errors in transcribing?

Line 157 –Was the questionnaire administrator for 9Q trained or qualified to conduct the questionnaire (i.e. must be a doctor with training in psychiatry)?

Line 186 – As HbA1c is the most important outcome data in this study, describe in detail how HbA1c level is measured (etc. whole blood? Fasting state? Point of care?)

Results:

Table 1 & 2: Please clarify the use of Fisher’s exact test for the comparison of median values between groups.

Table 3 & 5 – Please consider reporting the regression model significance and fit.

Discussion

Line 322 – Please check and correct the unit; whether mg/dL or mmol/L.

Professional proofreading is recommended to elevate writing quality of discussion section.

Conclusions

Conclusions were well-written.

General comments

Majority of the references are recent publications (published within 10 years).

---

## [Author Response · Author response to Decision Letter 1]

2 Oct 2023

Reviewer1: I have incorporated all of your suggestions into my revision. They were very helpful. Thank you.

Reviewer2: I have incorporated all of your suggestions into my revision. They were very helpful. Thank you.

---

## [Editor Report · Decision Letter 2]

10 Nov 2023

Prevalence and the Association between Clinical Factors and Diabetes-Related Distress (DRD) with Poor Glycemic Control in Patients with Type 2 Diabetes: A Northern Thai Cross-Sectional Study

PONE-D-23-07649R2

Dear Dr. yingchankul,

We’re pleased to inform you that your manuscript has been judged scientifically suitable for publication and will be formally accepted for publication once it meets all outstanding technical requirements.

Kind regards,

Shairyzah Ahmad Hisham, PhD.

Academic Editor

PLOS ONE

Additional Editor Comments (optional):

After 2 revisions, the manuscript is now accepted for publication. However, it is strongly recommended to send the manuscript for proofreading to ensure that all grammatical and formatting errors are addressed prior to publication. Congratulation!

---

## [Editor Report · Acceptance letter]

14 Nov 2023

PONE-D-23-07649R2 

Prevalence and the Association between Clinical Factors and Diabetes-Related Distress (DRD) with Poor Glycemic Control in Patients with Type 2 Diabetes: A Northern Thai Cross-Sectional Study 

Dear Dr. Yingchankul:

I'm pleased to inform you that your manuscript has been deemed suitable for publication in PLOS ONE. Congratulations! Your manuscript is now with our production department. 

Kind regards, 

on behalf of

Dr. Shairyzah Ahmad Hisham 

Academic Editor

PLOS ONE